# Formulation and evaluation of atorvastatin calcium trihydrate Form I tablets

**Karen Andrea Salazar-Barrantes**[1,2], **Ariadna Abdala-Saiz**[3], **José Roberto Vega-Baudrit**[2], **Mirtha Navarro-Hoyos**[4,5], **Andrea Mariela Araya-Sibaja**[2]*

1 Escuela de Química, Universidad de Costa Rica, San José, Costa Rica, 2 Laboratorio Nacional de Nanotecnología LANOTEC-CeNAT-CONARE, San José, Costa Rica, 3 Departamento de Investigación y Desarrollo, Calox de Costa Rica, San José, Costa Rica, 4 Laboratorio BIODESS, Escuela de Química, Universidad de Costa Rica, San José, Costa Rica, 5 Chemistry Department, Georgetown University, Washington, D.C., United States of America

* aaraya@cenat.ac.cr

**Data Availability Statement:** All relevant data are within the main manuscript and in the Supplementary material.

**Funding:** CeNAT Scholarship Program.

## Abstract

Solid forms transformations and new crystal structures of an active pharmaceutical ingredient (API) can occur due to various manufacturing process conditions, especially if the drug substance is formulated as a hydrate. The conversion between hydrate and anhydrate forms caused by changes in temperature and humidity must be evaluated because of the risk of dehydration and phase transitions during the manufacturing process. Differences in physicochemical, mechanical, and rheological properties have been observed between solid forms of the same API that can cause manufacturing and product-related issues. Atorvastatin calcium trihydrate (ACT) is a synthetic lipid-lowering agent that was discovered during Lipitor® (its anhydrous form) Phase 3 clinical trials after passing Phase I and II. This case highlights the importance of routinely performing solid form screenings because of the probability of finding new solid forms during the development and scale-up process. Therefore, in this contribution, ACT tablet formulation was performed and evaluated starting from the compatibility of 1:1 proportions of drug and the excipients microcrystalline cellulose 101 (MCC 101), calcium carbonate, lactose monohydrate, croscarmellose sodium, hydroxypropyl cellulose, magnesium stearate, and polysorbate 80. Then, 40 mg ACT tablets were prepared on a small pilot scale, and manufacturing process assessment was conducted by sampling process stages selected as critically prone to solid forms formation or phase transition. Final product quality was evaluated regarding weight variation, hardness, disintegration, dissolution, and assay tests. Powder X-ray diffraction (PXRD), Fourier transform infrared spectroscopy (FT-IR), differential scanning calorimetry (DSC), and thermogravimetric analysis (TGA) were applied to solid state evaluation. The starting raw material was confirmed to be ACT Form I. From the preformulation studies, PXRD, FT-IR and TGA analyses showed no interactions between ACT and excipients, while DSC results revealed a physical interaction with MCC 101, not considered an incompatibility. The effect of the tablet manufacturing process was achieved by amorphization, while some ACT long-range crystalline structure remained, as confirmed by PXRD, FT-IR and DSC. However, the tablets' quality parameters were found to be within the acceptable range of both the pharmacopeia

**Competing interests:** The authors have declared that no competing interests exist.

guidelines and manufacturer parameters regarding weight variation, hardness, disintegration, dissolution, and assay tests.

## Introduction

Hydrates are one subtype of solid solvates, where water molecules are incorporated into the crystal lattice compound. Hydrates are considered novel solid forms that account for one-third of drugs developed [1]. Like every crystalline active pharmaceutical ingredient (API), hydrates are prone to polymorphism, which refers to the formation of different crystal structures of the same chemical molecule. Each of these crystalline structures can possess different mechanical, thermal, physical, and chemical properties that affect solubility, bioavailability, hygroscopicity, melting point, stability, compressibility, and other characteristics that ultimately determine drug performance [2], product quality and safety.

It is well-known that solid forms transformations and new crystal structures of an API can arise due to various manufacturing process conditions. The conversion between hydrate and anhydrate forms caused by changes in temperature and humidity require special attention, particularly if the drug substance is formulated as a hydrate. Likewise, studying the interactions between various APIs and excipients during the pre-formulation stage is crucial, as the properties largely depend on the chosen excipients, their concentration, and API-excipient interactions [3]. Excipients are usually ignored by manufacturers, even though their solid form might affect stability, compressibility, wettability, and other crucial attributes in a continuous manufacturing operation [4]. Therefore, for the development of a pharmaceutical product of a hydrate drug, it is essential to understand the relationship between a particular solid form and its functional properties due to its impact on manufacturing processes and drug safety, effectiveness, and quality [4, 5]. To achieve this goal, the whole process needs to be studied to ensure there are no changes in the APIs crystal structure during its production.

Atorvastatin calcium trihydrate (ACT), chemically designated as [R-(R*,R*)]-2-(4-fluorophenyl)-$\beta,\delta$-dihydroxy-5-(1-methylethyl)-3-phenyl-4-[(phenylamino)carbonyl]-1H-pyrrole-1-heptanoic acid, calcium salt (2:1) trihydrate ($[C_{33}H_{34}FN_2O_5]_2Ca\cdot3H_2O$; MW 1209,42 g/mol; Fig 1), is a synthetic lipid-lowering agent that belongs to Biopharmaceutics Classification System (BCS) class II [6].

Obesity has become one of the leading global health issues, with its prevalence more than doubling over the last three decades. It now contributes to approximately 1.3 million deaths each year and is characterized by an abnormal or excessive buildup of lipids in the body [7].

**Fig 1. Chemical structure of atorvastatin calcium trihydrate (ACT).**

This significantly increases the risk of cardiovascular diseases, diabetes, and other metabolic disorders. A popular treatment for these disorders is ACT, an inhibitor of the 3-hydroxy-3-methylglutaryl coenzyme A (HMG-CoA) reductase, which catalyzes the conversion of HMG-CoA to mevalonate, an early and rate-limiting step in cholesterol biosynthesis [8, 9]. ACT is a hydrate form of the atorvastatin calcium anhydrous initially marketed by Pfizer under the name Lipitor®, holding the record of the best-selling medication in 2008 with a revenue of 12.4 billion dollars [8, 9]. It was indeed discovered during Phase 3 clinical trials after passing Phase I and II with the amorphous form [5]. This case highlights the importance of routinely perform solid form screenings because of the probability of finding new solid forms during the scale-up development process [5].

Considering ACT has a high incidence of polymorphs, hydrates, and solvates formation [10], the history of its parent compound and that to the best of our knowledge, there has not been reported a real tablet manufacturing process considering solid forms transformation for ACT form I. The present study aimed to contribute to the limited number of illustrative cases in the literature for this drug [11], starting with the drug-excipient interaction evaluation, the manufacturing process assessment and the final product quality test.

## Materials and methods

### Materials

ACT, excipients, including microcrystalline cellulose 101 (MCC 101), calcium carbonate (CaCO$_3$), lactose monohydrate (LM), croscarmellose sodium (CS), hydroxypropyl cellulose (HPC), magnesium stearate (MS), and polysorbate 80 (P80), API-excipient mixtures, and solids obtained at five points of the manufacturing process were kindly donated by Calox de Costa Rica (San José, Costa Rica).

### Drug-excipient compatibility study

To study the compatibilities between the API and excipients, solid samples were prepared by weighing 1:1 proportions of ACT and each excipient. Excipients tested and sample codes are shown in Table 1. They were mixed by manual agitation in amber glass bottles and stored at room temperature. The samples were analyzed through the PXRD, FT-IR, DSC and TGA techniques according to the setup described in the Solid-state characterization section.

### Tablet manufacturing process evaluation

0.8 kg of 40 mg ACT tablets were prepared according to the schematic process shown in Fig 2. Some specific conditions were not described here due to the confidentiality of the pharmaceutical industry involved in this study. Samplings were performed after ACT was subjected to

Table 1. Excipients mixed with ACT and their codes used in the compatibility study.

| Sample code | Excipient |
| --- | --- |
| T-1 | Microcrystalline cellulose 101 |
| T-2 | Calcium carbonate |
| T-3 | Lactose monohydrate |
| T-4 | Croscarmellose sodium |
| T-5 | Hydroxypropyl cellulose |
| T-6 | Magnesium stearate |
| T-7 | Polysorbate 80 |

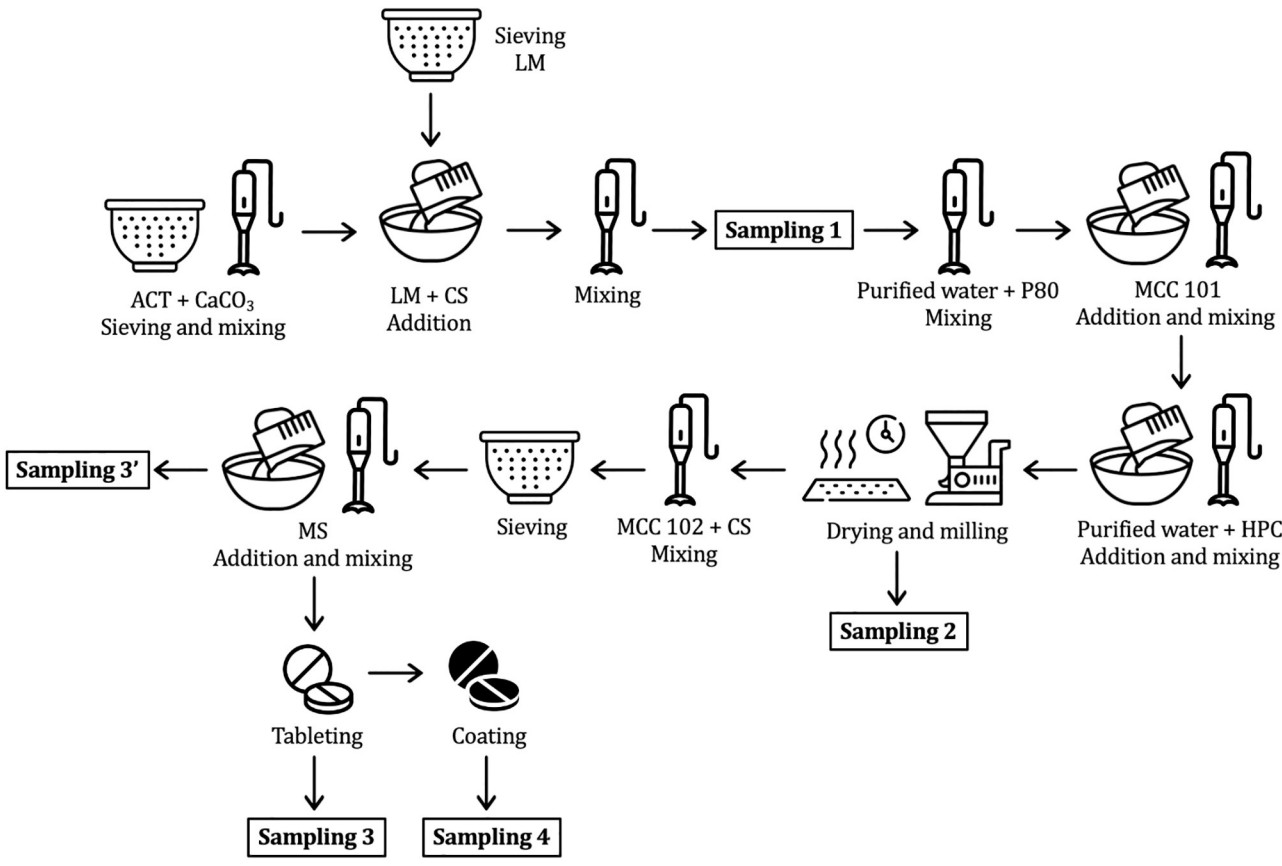

**Fig 2. Schematic representation of the manufacturing process.**

conditions considered critical, which are known to influence polymorphic formation or phase transition. Sampling 1 was taken after sieving, mixing and addition of excipients. Sampling 2 was obtained after two wet granulation processes, drying and grinding steps. Sampling 3' was withdrawn after mixing and sieving, while Sample 3 was obtained after compression of Sampling 3'. Finally, Sampling 4 came from film coating in which a solvent is applied by spray-atomization and fast drying. The collected samples were analyzed through the PXRD, FT-IR, DSC and TGA techniques according to the setup described in the Solid-state characterization section.

## Tablet quality test

In addition, tablets' physical and chemical tests were performed according to the standard procedures [12] as described below.

**Weight variation.**  The weight variation test was conducted according to the United States Pharmacopeia (USP) General Chapter 230A⟨905⟩ Uniformity of dosage units using an analytical digital balance Sartorius BCE323-1S (Sartorius, Germany) by weighting accurately ten tablets individually and calculating the average weight for the random samples.

**Hardness.**  The hardness test was performed following the USP General Chapter ⟨1217⟩ Tablet breaking force using an Erweka-TBH 125 durometer (Erweka, Germany), recording the results in kilopond. The average hardness was calculated.

**Disintegration.** The test was carried out as indicated in the USP ⟨701⟩ Disintegration General Chapter on six tablets using a Vanguard Pharmaceutical Machinery LIJ-1 disintegration test apparatus (Vanguard Pharmaceutical Machinery, USA).

**Assay.** ACT content for both assay and dissolution test were quantified using an HPLC Shimadzu Prominence-i LC-2030 (Shimadzu, Japan). The mobile phase was composed of acetonitrile, tetrahydrofuran, and 0.05 M ammonium citrate buffer pH 4.0 (27:20:53), a flow rate of 1.5 mL/min, 20 μL injection volume, detection at 244 nm. Chromatographic separation was performed in a Phenomenex Luna C-18 column (5 μm; 250 mm x 4.6 mm) (Phenomenex, USA) kept at 30 ˚C.

**Dissolution.** The dissolution of ACT tablets was evaluated according to the USP monograph for atorvastatin calcium tablets on a Teledyne Hanson CD14 dissolution test system (Hanson Research, Chatsworth, CA, USA) using the USP dissolution apparatus 2, in 900 mL of 0.05 M phosphate buffer as medium, and a stirring rate of 75 rpm for 15 min. Then, the solutions were filtered using a 0.45 μm membrane filter and analyzed by HPLC to determine drug concentration.

**Drug content determination.** ACT content in the tablets was determined following the USP monograph. In brief, a solution was prepared by weighing and transferring ten tablets of ACT to a volumetric flask. Around 50% of the flask was completed with acetonitrile and 0.05 M of citric acid solution adjusted to a pH of 7.4 with ammonium hydroxide (1:1). The sample was mechanically mixed until dissolved and diluted to volume. The solution was filtered using a 0.45 μm membrane filter and diluted to achieve a nominally equivalent to 0.1 mg/mL of atorvastatin concentration. The solution was analyzed by HPLC to determine drug concentration.

## Solid-state characterization

**Powder X-ray diffraction (PXRD).** PXRD patterns under ambient conditions were recorded using a Bruker D8 Advance ECO (Bruker, Karlsruhe, Germany) diffractometer equipped with the DIFFRAC.SUITE software and a Cu Kα source ($\lambda$ = 1.2–1.54 Å) operated at 1000W, with a 2θ range between 2.5˚ and 35˚ at 0.6˚/min.

**Fourier transform infrared spectroscopy (FT-IR).** FT-IR spectra of the samples were recorded on a Thermo Scientific Nicolet 6700 FT-IR (Thermo Fisher Scientific, Waltham, MA, USA) spectroscope equipped with a diamond Attenuated Total Reflectance (ATR) accessory. A small amount of the solid sample was placed directly on the ATR without prior preparation. The data was collected in the range of 4000 to 500 cm$^{-1}$ using the OMNIC software.

**Thermogravimetric analysis (TGA).** TGA analyses of the samples were performed on a TA Instruments model Q500 thermogravimetric analyzer (TA Instruments, New Castle, DE, USA). Approximately 8 mg of the solid sample was placed in a platinum crucible, and a temperature ramp of 10 ˚C/min was established with a temperature range from 25 to 1000 ˚C under a nitrogen atmosphere flow of 90 mL/min on the sample and a 10 mL/min on the balance.

**Differential scanning calorimetry (DSC).** The DSC curves were obtained using a DSC-Q200 (TA Instruments, New Castle, DE, USA) calorimeter. Approximately 3 mg of each solid sample were weighed into unsealed aluminum crucibles and placed in the DSC cell along with an empty aluminum crucible as a reference. The measurements were performed under an inert nitrogen atmosphere of 50 mL/min and a heating rate of 10 ˚C/min over a temperature range from 20 to 300 ˚C. Glass transition events ($T_g$) were determined according to ASTM D3418 Standard Method.

## Results and discussion

### Compatibility study

As shown in Table 1, seven relevant excipients were selected to complement the materials used in a previous compatibility study at the preformulation level [3]. The appropriate selection of excipients is crucial, as it determines the fate of the final dosage form [13]. Thermal analysis provides quick, preliminary insights into potential interactions, allowing a direct evaluation of binary mixtures without requiring time-consuming stress conditions [14]. However, thermal analysis cannot fully predict compatibility, as the final formulation may differ from the preformulation mixtures [3]. Moreover, while the International Council on Harmonization (ICH) and regulatory authorities provide guidelines for accelerated stability testing of finished dosage forms, the choice of accelerated conditions for compatibility studies is at the formulator's discretion [15].

### PXRD and FT-IR analyses

The first step in controlling drug solid forms transformation is to identify and characterize the starting raw material. The crystal structure of ACT has not been deposited in the Cambridge Structural Database (CSD) of the Cambridge Crystallographic Data Center (CCDC). Only one structure, atorvastatin calcium ethylene glycol solvate (IZOQIZ), is available in this database [9]. In fact, there is a scarcity and ambiguous information regarding descriptions of the crystalline forms of ACT in the literature, including patents reported up to date [16–18]. Some of these forms are summarized in Table 2.

**Table 2. Crystal forms of ACT already reported in the literature.**

| Crystal form | Description |
|---|---|
| Form I<br>Pfizer (Warner-Lambert Company) | Crystal structure not reported in CSD.<br>Triclinic (P1) according to International Centre for Diffraction Data (ICDD)<br>Powder Diffraction File, (PDF®) entry 00-062-1582 [17, 18]<br>*Crystal data*: $a$: 5.4568(2) Å $\alpha$: 76.801(3)°<br>$b$: 9.8887(4) Å $\beta$: 99.177(5)°<br>$c$: 30.3091(9) Å $\gamma$: 105.318(5)°<br>$V$: 1527.1(1) Å³<br>$Z$: 1<br>Triclinic (P1) [18]<br>*Crystal data*: $a$: 5.44731(4) $\alpha$: 95.859(3)<br>$b$: 9.88858(16) $\beta$: 94.211(1)<br>$c$: 29.5925(10) Å $\gamma$: 105.2790(1)°<br>$V$: 1521.277(10) Å³<br>$Z$: 1 |
| Forms V, X and XV<br>Pfizer (Warner-Lambert Company) | Crystal structures not reported [19]<br>No cell parameters have been reported |
| Form VIII<br>Teva Pharmaceutical Industries Ltd. | Monoclinic unit cell [20]<br>$a$: 18.55–18.7 Å $\alpha$: 76.801(3)°<br>$b$: 5.52–5.53 Å $\beta$: 97.5–99.5°<br>$c$: 31.0–31.2 Å |
| Form VI<br>Morepen Laboratories Ltd. | Crystal structure not reported [21]<br>No cell parameters have been reported |
| Forms<br>M-2, M-3 and M-4<br>Morepen Laboratories Ltd. | Crystal structure not reported [22]<br>No cell parameters have been reported |
| Form 3<br>An et al. | Crystal structure not reported [23]<br>No cell parameters have been reported |

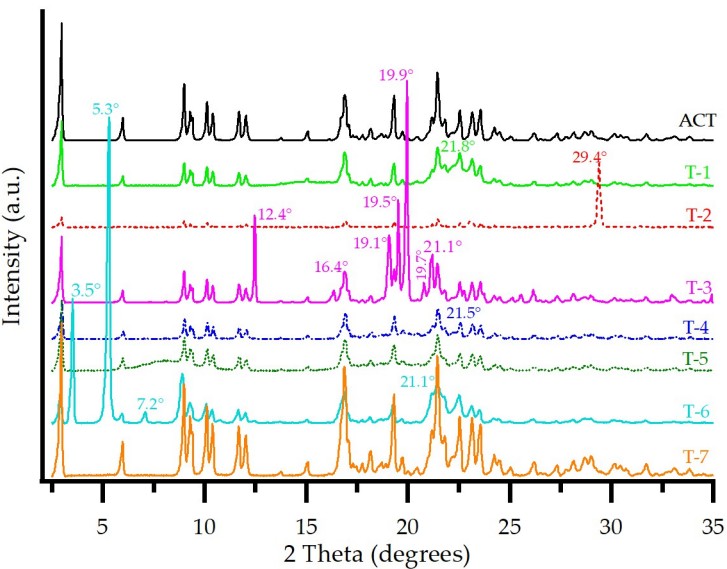

**Fig 3. PXRD patterns of ACT and each prepared ACT-excipient mixture.**

The ACT diffractogram shown in Fig 3 displays well-defined and intense reflections, which were consistent with the characteristic diffraction peaks of Form I for ACT described in U.S. patent 5,969,156 granted to Warner-Lambert (now Pfizer) in 1999 [16] and in the literature [18], including the PDF entry 00-062-1582 results, based on synchrotron data [17].

Fig 3 also shows the PXRD patterns of ACT compared to those of the ACT-excipient mixtures. T-1 and T-5 presented relatively broad and diffuse halos instead of well-defined peaks, possibly because of the lack of long-range three-dimensional order in these two amorphous solids [24]. However, the characteristic peaks of ACT, although decreasing in intensity, were maintained, indicating no interaction between the drug and the excipients.

For T-4, no interaction was evident because the API's characteristic peaks were distinguishable, and no new peaks were observed. This mixture exhibited a broad halo between 20.0 and 25.0° 2θ approximately, attributed to the CS excipient due to its characteristic semi-crystalline form, which is observed in different cellulose polymers [25].

The T-2, T-3, and T-6 mixtures showed a decrease in the reflection intensities for ACT. This can be attributed to these excipients masking the 2θ region [26]. However, the reflections of the API matched those of the API in the mixture, implying no interactions. In addition, the appearance of new peaks corresponded to the characteristic ones of the excipients used for each mixture. Through the software, it was possible to identify the main reflections. These are indicated for each case in Fig 3. Finally, T-7 showed no interaction between the drug and the P80 excipient.

The FT-IR spectra of ACT and each ACT-excipient mixture are presented in Fig 4. For ACT, the spectrum reveals a broadband (3700–2800 cm$^{-1}$) [3], which is overlapped by the asymmetric O-H stretching (3250 cm$^{-1}$) [27, 28], including the N-H stretching (3360 cm$^{-1}$) [28, 29]. 1650 cm$^{-1}$ corresponds to the C = O stretching of the amide carbonyl and 1580 cm$^{-1}$ to the C = O stretching of the carboxylate [3]. The peak at 1560 cm$^{-1}$ is characteristic of the C-N stretching, while the bands at 1510 cm$^{-1}$ and 1436 cm$^{-1}$ are due to the deformation of N-H and C-H, respectively [3]. The 1312 cm$^{-1}$ band is attributed to the methyl and methylene deformation vibrations, and the peak at 1216 cm$^{-1}$ to the C-F stretching [30, 31]. The bands at 848, 819, and 750 cm$^{-1}$ are an indication of the bending of out of the plane aromatic C-H

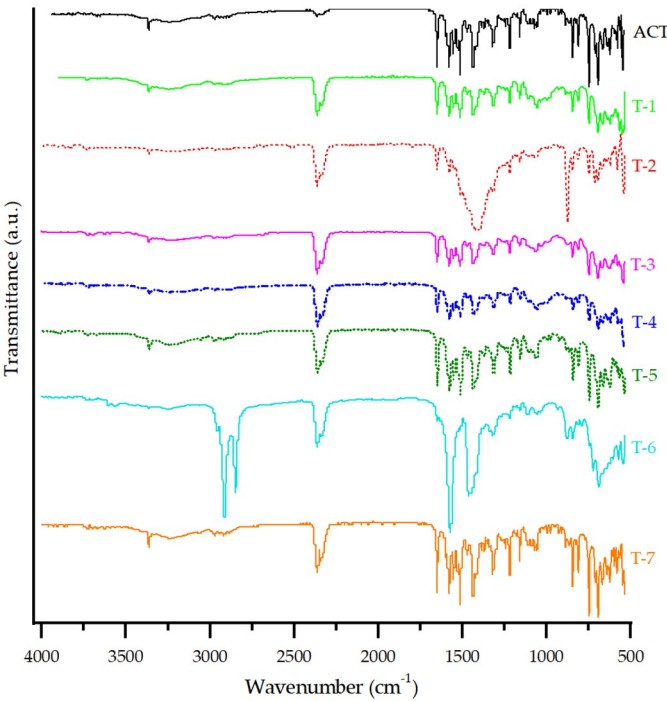

**Fig 4. FT-IR spectra of ACT and each prepared ACT-excipient mixture.**

bonds. Finally, the peak at 692 cm$^{-1}$ is characteristic of the C-H bonds bending deformation [31].

For T-1, T-3, T-4, T-5, and T-7, the characteristic peaks for ACT alone remained the same or shifted within a range of ± 5 cm$^{-1}$, suggesting that no interactions occurred [3]. T-2 exhibited characteristic signals of ACT. However, some peaks were not visible due to the overlapping of the characteristic bands of CaCO$_3$, with no indication of a drug-excipient interaction [3].

Finally, T-6 showed twin peaks at 1577 and 1450 cm$^{-1}$ due to asymmetric and symmetric carboxylate stretching vibrations (COO–), respectively. The peaks at 2915 and 2850 cm$^{-1}$ are assigned to the C–H stretching vibration [32]. These peaks overlapped with the drug characteristic bands without a clear indication of interaction.

**Thermal analysis.** The DSC curves of ACT and ACT-excipient mixtures are available in Fig 5. The temperature range was slightly different for each sample according to the temperature range that covers thermal events not only for ACT but also for the excipients involved in the study.

ACT showed two endothermic peaks, the first event at approximately 80–120 ˚C was found to be related to the loss of water by evaporation, then it was followed by another event at 155.4 ˚C, which corresponds to the melting point of ACT Form I reported in the literature [30, 33, 34]. The curve in the range 190–210 ˚C shows its decomposition [30].

For the T-1 DSC curve, the excipient promoted the appearance of a shorter melting peak for ACT and a slight displacement to a lower temperature, indicating a physical interaction, as the disappearance of the melting peak for ACT did not occur [35].

DSC curves for T-2, T-4, and T-7 maintained the melting peak of the API. Its displacement is insignificant compared to the pure drug, suggesting no thermal-induced interactions and, consequently, compatibility between both. Other peaks may be associated with

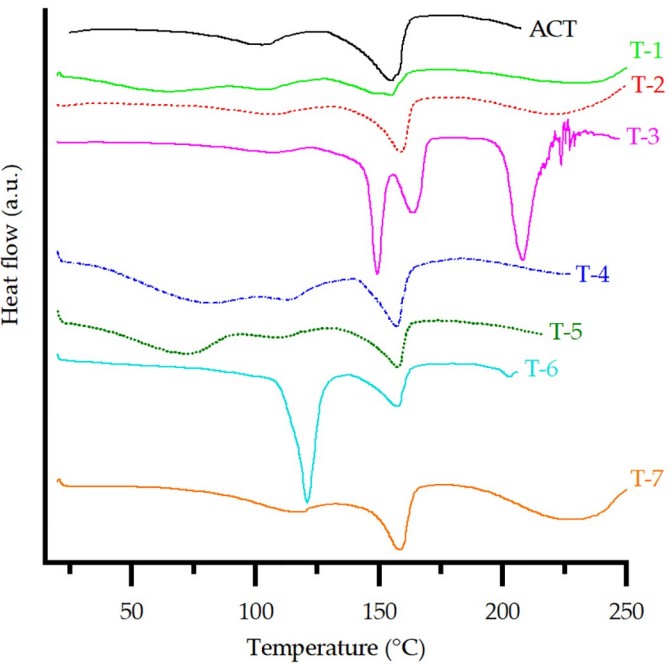

**Fig 5. DSC curves of ACT of ACT and each prepared ACT-excipient mixture.**

physicochemical transformations of the excipients [3]. For example, the decomposition of $CaCO_3$ to CaO starts at higher temperatures compared to ACT [36].

T-3 showed an endothermic peak corresponding to LM crystalline dehydration and melting at 149.27 ˚C and 208.23 ˚C, respectively. In addition, the corresponding melting peak for ACT at 163.88 ˚C was found. Lactose melts at 145 ˚C, which is lower than the melting temperature of ACT, thus promoting its solubilization in lactose [35] and affecting the intensity and width of the peak for the drug due to a solubility interaction and not a degradation of ACT. Therefore, it is not categorized as an incompatibility.

For T-5, the endothermic peaks for water loss and ACT melting maintained their shapes and slightly shifted their values, confirming good physical compatibility. T-6 exhibited the reproducibility of endothermic events for each compound separately, indicating compatibility between the drug and excipient. Small variations in the melting temperature of the drug are due to the decrease in individual purity without indicating incompatibility [37]. The disappearance of the endothermic event related to the loss of water from ACT at 101.88 ˚C is due to the superposition of the endothermic event of MS, related to the fusion of magnesium palmitate, as stearic acid and palmitic acid are present in its composition (impurities commonly found in commercial batches) [37].

Fig 6 shows the TGA curve (solid line) and the TGA derivative (dotted line) for ACT in which three mass loss steps were observed above 150 ˚C, specifically at 63, 97 and 115 ˚C. Only one thermal event was observed in the DSC curve near 100 ˚C. The first and second steps exhibited a loss of mass of around 1.5%, and the third step a loss of 1.7% for a total of 4.7%. Considering ACT and water molecular masses of 1209.39 g/mol and 18.015 g/mol, respectively, these percentages corresponded to one water molecule each for a total of three molecules. These results align with the reports in the literature that ACT has a percentage of bound trihydrate equal to 4.46%, equivalent to 2.8 water molecules in total [29, 34]. Thus, our result confirmed that the sample is effectively a trihydrate. Solid-state NMR studies reported in the

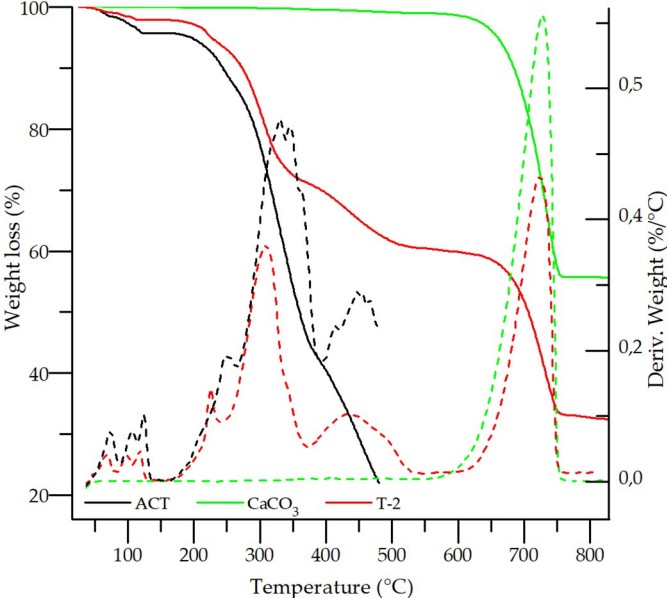

**Fig 6. TGA curves of ACT, CaCO$_3$ and T-2 sample.** Dotted lines represent TG derivatives.

literature explain that two water molecules bridge two divalent calcium ions in ACT, which explains the high temperature at which these two molecules leave the crystal (above 100 ˚C). Meanwhile, the ACT water loss signal shown below 100 ˚C is most likely related to the loss of the water molecule bound to a single calcium ion [34].

Finally, the start of the decomposition of ACT is observed around 190–200 ˚C in the TGA curve in Fig 6.

TGA curves of CaCO$_3$ and the T-2 sample are also presented in Fig 6. The binary mixture showed a lower rate of mass loss compared to the pure drug at high temperatures. Additionally, the decomposition of the drug-excipient mixture was initiated at a higher temperature than in the pure API, showing an improvement in the stability of ACT mixed with CaCO$_3$. The three water molecules of ACT were lost by evaporation at 220.99 ˚C, 41.65 ˚C higher, confirming the previously discussed findings.

The TGA curves (S1 Fig) correspond to the other binary mixtures; the ACT thermal stability was found to be unchanged by the excipients tested. A slight improvement in thermal stability was observed, without signs of thermally induced interactions between the ACT and excipients.

## Tablet manufacturing process evaluation

During pharmaceutical manufacturing steps, several phase transformations may occur in the API [4]. Sampling points were chosen according to conditions considered critical, known as process-induced transformations (PITs) for an API. PITs occur when mechanical or thermal stress are applied to a system during processing. Critical conditions for PITs include moisture exposure during wet granulations and film coating, temperature changes during drying, and shear forces during milling, sieving, mixing, and compression. These processes include the production of amorphous regions, polymorphic transformations, dehydration of crystalline hydrates and hydration of anhydrous crystals [38].

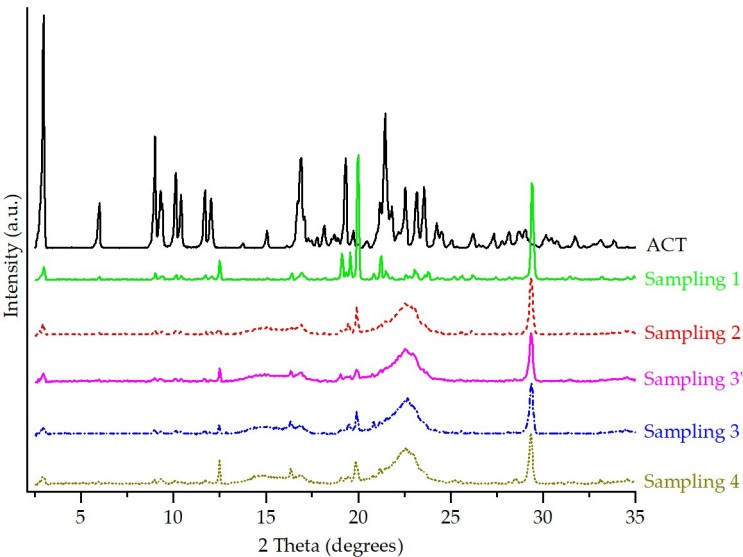

**Fig 7. PXRD patterns of ACT and samplings from the 40 mg tablet manufacturing process.**

**PXRD and FT-IR analyses.**    Fig 7 shows the PXRD patterns of ACT compared with the samplings 1 to 4 (see scheme in Fig 2) diffractograms.

A polymorphic transition may be triggered by using the simplest processing method (e.g., mixing) [39]. Therefore, Sampling 1 was taken after the first manufacturing steps, including sieving, mixing and the addition of excipients. For Sampling 1–the initial prepared mixture–reflections related to LM and $CaCO_3$ were observed–both determined from the signals of the binary mixtures (Fig 3). The decrease in the intensity of the characteristic peaks for the API was attributed to these excipients masking the 2θ region [26], as mentioned previously. A decrease in the peak intensity of the reflections may also imply a smaller change in the crystal lattice, such as the introduction of a slight disorder or change in lattice parameters [40]. During milling, the API and excipients are exposed to large shearing forces and a temperature increase; therefore, transformations are likely to occur during this phase. In most cases, the material becomes structurally disordered, facilitating the escape of the hydrate water [41].

Sampling 2 was obtained after two wet granulation processes, drying and grinding steps. In the wet granulation processes, particles are aggregated to larger granules by the addition of an aqueous binding agent. The mixture is exposed to water, high humidity and temperature, and mechanical activation that could cause solid-state transformation [42]. At this step of the fabrication process, the first and second wet granulations involve purified water, along with excipients such as MCC 101 and HPC. The latter is characterized by containing water, which can escape into the headspace of a closed container, resulting in significant relative humidity. During these, water acts as a plasticizer for locally disordered regions in the crystal or partial amorphous structure, which may have been obtained during the initial milling phase, leading to a significant increase in molecular mobility [43]. This water, located at the interface between the API and the excipient, can also plasticize them to facilitate potential interactions or phase changes. Sufficient plasticization may even cause API molecules to migrate into the structure of a plasticized polymer, forming a ternary solution of drug-excipient-water in an amorphous solid state [43]. PXRD diffractogram of Sampling 2 showed that a few reflections related to ACT, LM, and $CaCO_3$ were maintained with a lower intensity.

During the drying phase, which is also part of Sampling 2, hydrates tend to lose the water from the crystal, henceforth forming dehydrated hydrates. This phenomenon leads to the creation of activated sites in the crystal, which have a propensity to either reabsorb the solvent molecules or to associate themselves with other moieties available, such as from the excipients that are part of this mixture [44]. Furthermore, the crystal lattice of a hydrate could collapse, leading to the formation of an amorphous phase. Although slow dehydration generally results in a crystalline phase, rapid removal of water can induce amorphization [45]. Drying is commonly performed as fluid bed operation or tray drying [42]. Previous investigations for a proprietary development compound that has seven different polymorphs/hydrates showed a variation in the solid phases found in the fluid-bed dried with the amount of water used during high shear granulation and the moisture content of the dried granules [46]. Also, it has been demonstrated that milling induces a partial structural disorder due to the weakness of the hydrogen bond in the crystal lattice [47]. The PXRD pattern of Sampling 2 compared to Sampling 1 exhibits the appearance of a halo pattern due to the decrease in the crystallinity of the API, suggesting its amorphization [4, 27, 48–50].

The energy and heat generated during the milling step are not controlled, but they can induce various solid-state changes, where further impact can be produced from the existence of excipients and water [42]. Impact mills and fluid-energy mills are both widely utilized in the pharmaceutical industry [46]. Different phase transitions of organic compounds during this process have been demonstrated in the literature [51–53]. Sampling 3', corresponding to the mixing with extragranular excipients step, showed significant intensities decrease of the peaks previously assigned for LM and $CaCO_3$. The result seems to indicate that ACT is no longer in a completely crystalline form when processed with polymers, but rather, an amorphization of it occurred, as reported in the literature. Lemsi et al. (2017) investigated the presence of solid-solid interaction with heating between atorvastatin and a polymeric matrix of vinylpyrrolidone and hydroxypropyl methylcellulose. Their thermal analyses results showed a complete dissolution of atorvastatin in these polymers below the melting point of the drug [27]. According to Mártha et al. (2013), the polyvinylpyrrolidone (PVP) polymers (PVP C30 and PVP K25) caused the amorphization of the meloxicam drug during a co-grinding process [54]. Also, Mura et al. (2002) found that the presence of PVP during the co-grinding process with the glisentide drug favored its amorphization [55].

Before tablet compression, granules are directly blended with lubricants and/or other excipients. Although this process has been considered previously as a minimal risk of phase transitions, Sampling 3' was considered. At this stage, the granulated solid is often subjected to milling and the intensity and energy exerted on it is generally lower, but the presence of excipients makes the detection of phase transitions difficult [46].

During tableting, granulations may be subject to compression forces as high as 40 kN with a few milliseconds, where defects such as cracks or an increase in the amorphous content may occur [42, 46]. Sampling 3 was taken after this energy impact to study the solid phase changes in the API via the solid-state mechanism [46]. Increased dehydration rate has also been observed for cyclophosphamide monohydrate and creatine monohydrate tablets [42]. Finally, the film coating process was applied to obtain Sampling 4. A film coating is often applied as an aqueous or solvent-based polymer system in coating pans or a fluid bed. This process involves the application of a spray-atomization technique, where also rapid drying takes place consequently to evaporate the solvent [42, 46]. Local water activity on the tablet core-film interface can be high enough to introduce hydrate formation, affecting the desired behavior of the final dosage form [46]. Samplings 3 and 4 exhibited a persistence of some characteristic ACT reflections with a very low intensity. Reflections for $CaCO_3$ and LM were also maintained with a higher intensity than Sampling 3' for the latter excipient. This may be due to a change in

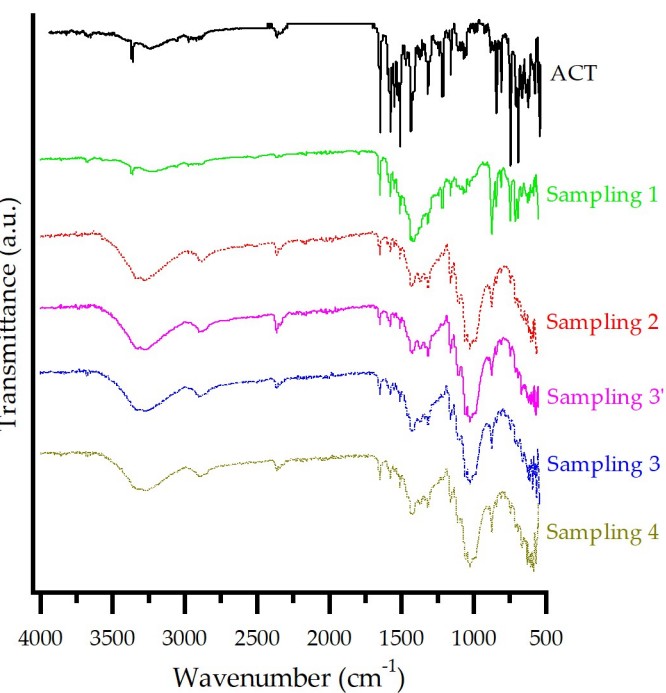

**Fig 8. FT-IR spectrum of ACT and samplings from the 40 mg tablet manufacturing process.**

particle size distribution by the resulting mixture during the addition of extra-granular excipients. For Sampling 4, it was found that the interaction between the core material and the coating liquid was minimal during film coating. Because of a highly efficient air exchange, there could have been only a short time between the impingement of coating liquid onto the tablet surface and the subsequent solvent evaporation [46].

The FT-IR spectra of ACT, as well as the samplings taken during the manufacturing process, are presented in Fig 8. Sampling 1 still shows characteristic signals previously assigned for ACT. Meanwhile, $CaCO_3$ caused an overlap of characteristic excipient bands and ACT, such as the signal within the range of 1490 cm$^{-1}$ to 1300 cm$^{-1}$. Modifications in Sampling 1 due to LM or CS were not evident during this step. Therefore, there were no changes in the integrity of the API.

Sampling 2 exhibits slight negative differences in the wavenumber of characteristic bands such as 1650 cm$^{-1}$ (asymmetric stretching of the C = O from the secondary amide) and 1580 cm$^{-1}$ (C = O symmetric stretching). These changes can be attributed to the relative amorphization disorder of molecules, resulting in a broader distribution of bond lengths and energies compared to their crystalline counterparts [27]. Also, hydrogen bond related signals correspond to wider peaks, such as the one at 3360 cm$^{-1}$ (N–H stretching) and 3250 cm$^{-1}$ (asymmetric O–H stretching) [27]. Therefore, ACT shows some degree of H-bonding with HPC through the secondary amide groups, a molecular site associated with the conformational changes occurring predominantly in the amorphous state [27, 56, 57]. The presence of this H-bonding has also been related to the physical stability of amorphous dispersions [56]. Consequently, the effect of grinding changes in the molecular structure of ACT was evidenced, which reflects a possible amorphization [27, 31].

Within the same broad absorption band mentioned before (3300–3600) cm$^{-1}$, assigned to O–H stretching, the presence of P80 free hydroxyl groups [58] could also be established,

contributing to the FT-IR spectrum of Sampling 2. The strong reduction of intensity in the aromatic C–C stretching bands (1550–1450) cm$^{-1}$ for both complexes might be related to vibrational restrictions imposed upon complexation [27]. These results indicate a physical interaction via hydrogen bonding or complexation between ACT and polymers [59].

For Sampling 3', the assigned signals for the previous sampling had no shifts and even the appearance of the broad signal at approximately 1030 cm$^{-1}$ remained, indicating no changes in the amorphization obtained previously for ACT. Samplings 3 and 4 showed again no significant change in the infrared spectra. It was observed that once the amorphization of the ACT was obtained, it remained constant in the manufacturing process.

**Thermal behavior.** Fig 9 shows the DSC curves obtained for ACT and the samplings studied samplings. As previously mentioned, DSC curves showed different temperature ranges to cover main thermal events not only for ACT but also for the excipients involved in the study. Sampling 1 exhibits three endothermic peaks. The first one is related to the loss of water due to dehydration of the ACT. However, the endothermic event related to the melting point of the API is not present, suggesting the amorphization of the API [30] and its solubility interaction with LM, as discussed earlier [35]. The disappearance of the endothermic event for ACT can be explained by the dilution effect of the polymer [60].

The DSC curves of all the samplings during the manufacturing process presented a broad endotherm in a temperature range of approximately 40 to 130 ˚C, corresponding to the release of adsorbed and structural water from the API [27]. Because of the broad and prominent shouldering of the signals, a molecular reorganization could be the reason for the analysis time (10 ˚C /min) [6].

Sampling 2 shows a similar behavior. Excipients such as HPC in this mixture exhibited a broad peak corresponding to dehydration, between 40 and 97.5 ˚C, and did not show characteristic peaks of ACT, indicating the possible complete dissolution of it in this polymer. Many

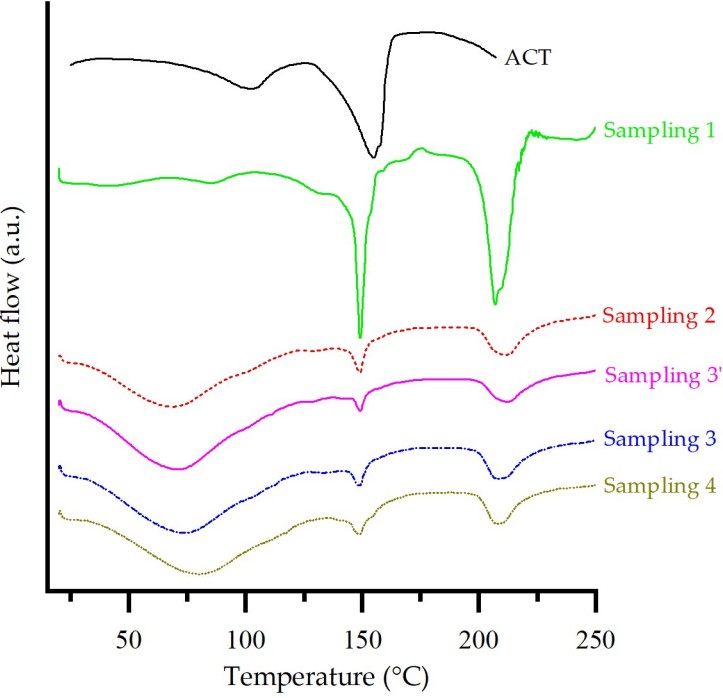

**Fig 9. DSC thermograms for ACT and samplings from the 40 mg tablet manufacturing process.**

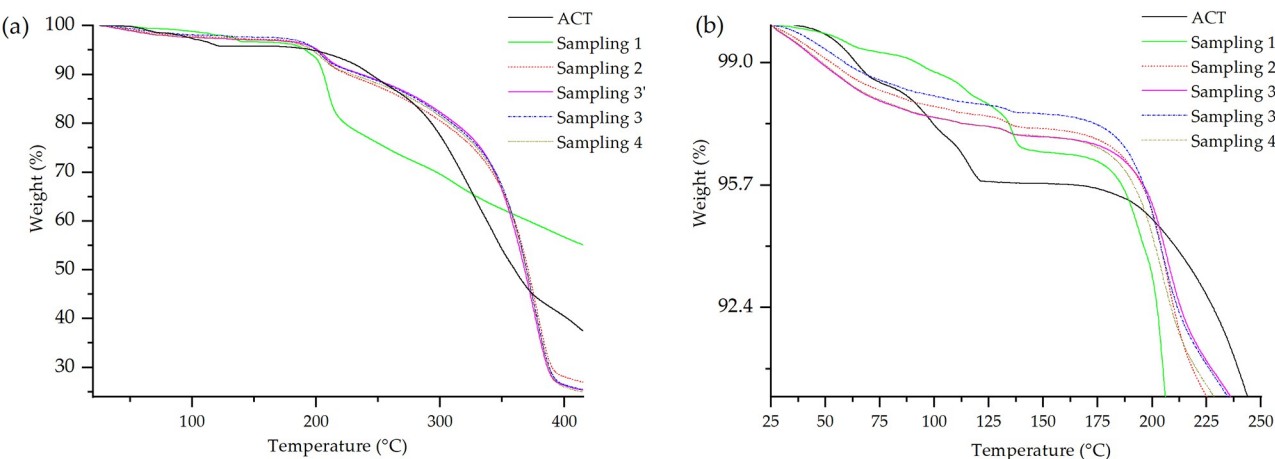

**Fig 10. TGA curves for the samplings collected during the manufacturing process of the 40 mg tablets.** (a)–results up to 410 ˚C; (b)–results up to 250 ˚C.

studies have reported that the miscibility of the drug with the polymer is directly related to the stabilization of an amorphous drug against crystallization [27].

Because of the high presence of water observed in Samplings 1 and 2, it is established that the polymers prevent the recrystallization of the API despite the role of water as a reactant to enhance degradation and mobility (plasticization) [27]. No changes were observed in the DSC curves obtained when comparing Sampling 2 with Sampling 3'. Samplings 3 and 4 showed very similar endothermic events.

The mixture of excipients decreases the purity of each of the prepared mixtures, which is shown as a broadening of the endotherms. Therefore, it is ruled out that the purity of the compounds is a factor that affects the melting point of ACT [57].

Although amorphization was evident by PXRD data, it was further complemented and confirmed by DSC experiments, as described below. S2 Fig revealed a glass transition temperature ($T_g$) for samplings 2 to 4 observed between 145 and 155 ˚C in agreement with ACT reports on $T_g$ in literature [27], thus confirming API's amorphization.

The thermal stability of the samplings was studied by comparing their weight loss with the pure drug ACT through the TGA curves shown in Fig 10. Sampling 1 shows the dehydration of ACT, although the weight loss is around 0.9% less than that of ACT as a pure drug. In this case, the excipients in the initial blend (CaCO$_3$, LM, and CS) provide stability to the drug. The first relatively smaller step change was most likely due to the loss of loosely bound water. Subsequently, second and third step changes were due to the loss of water of hydration. The fall in the baseline temperature of 175 ˚C was due to its degradation [6].

Samplings 2, 3', 3 and 4 showed only one broad step weight change in the temperature range of 30–160 ˚C, followed by degradation at a temperature of about 175 ˚C. The loss of crystallization water from ACT dominates the behavior of the mixture. The results indicate that once the crystallization water is lost from ACT, the thermal stability of the mixtures is reduced. This is due to the disorganization of the ACT unit cell (loss of crystallization water), which favors the transfer of heat to the core of the system and the subsequent thermal decomposition [61].

TGA curves of Samplings 3', 3, and 4 showed the same behavior as Sampling 2, apropos the absence of proper dehydration events of ACT. In the case of Sampling 3', its weight loss is higher. This may be due to the addition of the extra-granular inputs used before the tableting.

Sampling 3 shows a lower weight loss than samplings 3' and 2 during thermogravimetric analysis until reaching decomposition. In this case, the tableting is performed once the MCC 102 is added as an extragranular excipient. This mainly exhibits plastic deformation, where the particles deform like plasticine under compression, promoting the increase of intermolecular binding forces between excipients and API, mainly van der Waals, reducing the space between particles and resulting in the formation of a common surface by deformation [62].

The compaction force at this point can shorten the distance of the particles that make up the tablet and easily induce thermal conductivity for molecular interaction [63]. These bonds and short distances may promote thermal stability of the manufactured tablet, affecting the analysis. Therefore, the stability and short bonding distances will require more energy to break, causing the weight loss to be lower during its decomposition.

Finally, Sampling 4 was found to have a behavior similar to Sampling 3'. The peak temperatures of all dehydration-based endothermic events decreased with increasing grinding time, and the weight loss of the lattice water decreased, whereas that of the surface-adsorbed and channel water increased [64].

## Tablet quality tests

The weight variation, hardness, disintegration, dissolution, and assay for the 40 mg ACT tablet batch obtained by direct compression were determined and summarized in Table 3.

Acceptance limit criteria were estimated to range from 597 to 631 mg for weight variation and from 137.3 to 172.6 N for hardness. The results obtained were within these limits for both parameters. On the other hand, disintegration time met the defined parameter for this product, which was 10 minutes. Further, the drug content assay was within 94.5% and 105.0% limits as recommended by the USP criterion for this drug. In turn, the drug dissolved no less than 80% of the labeled amount in 15 minutes to accomplish the acceptance criteria. In sum, the physical characteristics were found to be within the parameters of the current USP atorvastatin calcium tablets monograph [12] and the established by the manufacturer.

## Conclusions

Compatibility studies of ACT with excipients for solid dosage forms development showed no interactions, as confirmed by PXRD and FT-IR. However, the overlapping of characteristic bands in the spectra of calcium carbonate and magnesium stearate did not provide a clear indication of drug-excipient interaction. DSC analyses demonstrated a physical interaction between ACT and microcrystalline cellulose 101, likely heat-induced, with a lower melting temperature.

During tablet manufacturing, ACT Form I was partially amorphized through milling, wet granulation, drying, blending, and compression, while some long-range crystalline structure remained. PXRD, FT-IR, and DSC minor changes in the crystal lattice of ACT during milling, without changes in its chemical structure. After wet granulations and drying, ACT formed an

Table 3. Quality parameter evaluated on the 40 mg ACT tablet batch prepared.

| Physical and chemical characteristics | 40 mg ACT tablet batch |
|---|---|
| Weight (mg) | 621 |
| Hardness (N) | 157.8 |
| Disintegration (s) | 40 |
| Dissolution (%) | 98 |
| Assay (%) | 98.90 |

amorphous solid, which was retained through blending, compression, and coating. TGA confirmed the disorganization of the ACT unit cell, which favored thermal decomposition but did not affect the quality of the final product, which met USP and manufacturer acceptance criteria, including drug dissolution.

The outcomes presented highlight the importance of excipient choice, conditions, and unit operations in tablet manufacturing. Therefore, during preformulation studies, drug-excipient mixtures should be evaluated using complementary techniques and identifying steps where mechanical activation, temperature, humidity, or solvent may promote a phase transition or solid-state changes. Furthermore, these results contribute to enriching the scarcity of studies on polymorphic stability throughout real manufacturing processes.

## Supporting information

**S1 Fig. TGA curves of ACT, excipients, and mixtures: (a) T-1, (b) T-3, (c) T-4, (d) T-5, (e) T-6, (f) T-7.**
(PDF)

**S2 Fig. DSC thermograms for ACT and samplings from the 40 mg tablet manufacturing process used to determine $T_g$ of ACT.**
(PDF)

## Acknowledgments

Authors thank the National Center for High Technology of Costa Rica CeNAT-CONARE, the National Laboratory of Nanotechnology (LANOTEC), CALOX de Costa Rica and the University of Costa Rica.

## Author Contributions

**Conceptualization:** Karen Andrea Salazar-Barrantes, Ariadna Abdala-Saiz, Mirtha Navarro-Hoyos, Andrea Mariela Araya-Sibaja.

**Data curation:** Karen Andrea Salazar-Barrantes, Ariadna Abdala-Saiz, Andrea Mariela Araya-Sibaja.

**Formal analysis:** Karen Andrea Salazar-Barrantes, Ariadna Abdala-Saiz, Mirtha Navarro-Hoyos, Andrea Mariela Araya-Sibaja.

**Funding acquisition:** José Roberto Vega-Baudrit, Mirtha Navarro-Hoyos, Andrea Mariela Araya-Sibaja.

**Investigation:** Karen Andrea Salazar-Barrantes, Ariadna Abdala-Saiz, Mirtha Navarro-Hoyos, Andrea Mariela Araya-Sibaja.

**Methodology:** Karen Andrea Salazar-Barrantes, Ariadna Abdala-Saiz, Mirtha Navarro-Hoyos, Andrea Mariela Araya-Sibaja.

**Project administration:** José Roberto Vega-Baudrit, Andrea Mariela Araya-Sibaja.

**Resources:** José Roberto Vega-Baudrit, Mirtha Navarro-Hoyos, Andrea Mariela Araya-Sibaja.

**Supervision:** José Roberto Vega-Baudrit, Mirtha Navarro-Hoyos, Andrea Mariela Araya-Sibaja.

**Writing – original draft:** Karen Andrea Salazar-Barrantes, Mirtha Navarro-Hoyos, Andrea Mariela Araya-Sibaja.

**Writing – review & editing:** Karen Andrea Salazar-Barrantes, Ariadna Abdala-Saiz, José Roberto Vega-Baudrit, Mirtha Navarro-Hoyos, Andrea Mariela Araya-Sibaja.

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
