## [Decision Letter · Decision Letter 0]

24 Sep 2024

PONE-D-24-20013Formulation and Evaluation of Atorvastatin Calcium Trihydrate Form I TabletsPLOS ONE

Dear Dr. Araya-Sibaja,

Thank you for submitting your manuscript to PLOS ONE. After careful consideration, we feel that it has merit but does not fully meet PLOS ONE’s publication criteria as it currently stands. Therefore, we invite you to submit a revised version of the manuscript that addresses the points raised during the review process.

We look forward to receiving your revised manuscript.

Kind regards,

Amjad Khan, Pharm-D; PhD

Academic Editor

PLOS ONE

Journal Requirements:

3. Thank you for stating the following financial disclosure: “CeNAT Scholarship Program”

4. In the online submission form, you indicated that “Data will be made available on request.”

All PLOS journals now require all data underlying the findings described in their manuscript to be freely available to other researchers, either 1. In a public repository, 2. Within the manuscript itself, or 3. Uploaded as supplementary information. This policy applies to all data except where public deposition would breach compliance with the protocol approved by your research ethics board. If your data cannot be made publicly available for ethical or legal reasons (e.g., public availability would compromise patient privacy), please explain your reasons on resubmission and your exemption request will be escalated for approval.

Reviewers' comments:

Reviewer's Responses to Questions

**Comments to the Author**

1. Is the manuscript technically sound, and do the data support the conclusions?

Reviewer #1: Yes

2. Has the statistical analysis been performed appropriately and rigorously? 

Reviewer #1: Yes

3. Have the authors made all data underlying the findings in their manuscript fully available?

Reviewer #1: Yes

4. Is the manuscript presented in an intelligible fashion and written in standard English?

Reviewer #1: Yes

5. Review Comments to the Author

Reviewer #1: After reviewing the paper, I am pleased to report that I consider the article is well written and presents a solid piece of research. However, I have identified some areas that could be improved to further strengthen the quality of the document.

Force units

One of the main observations is that in some sections the International System (SI) is not used for force units. I suggest you review and update these units to conform to international standards. This will help improve the coherence and clarity of the manuscript.

Use of the % sign

Additionally, I have noticed that in some cases the % sign is not separated from the number by a single space. I recommend that you review and correct these cases to maintain a consistent format throughout the text.

Viewing the figures

On the other hand, I have observed that the difference between the dark colors in some of the figures could be further reinforced using different types of plotting. This would make it easier to view and understand the data presented.

Filter Information

I would also like to point out that when the use of filters is mentioned in the text, no information is given about the specific material of the filters used. This detail may be crucial for the reproducibility of future research based on this study. I suggest you include this information in the corresponding section.

Thanks for counting on me

6. PLOS authors have the option to publish the peer review history of their article (what does this mean?). If published, this will include your full peer review and any attached files.

Reviewer #1: No

---

## [Author Response · Author response to Decision Letter 0]

4 Oct 2024

Academic editor:

The manuscript format has been modified and now meets PLOS ONE’s style requirements, including article title, corresponding authorship, levels 1 to 3 headings and file naming.

Funding Information’ and ‘Financial Disclosure’ have been fixed. A grant number is not available (NA) for this funding. Since the National Center for High Technology of Costa Rica (CeNAT-CONARE) Scholarship Program was for the financial support awarded to the student during the research period the state "The funders had no role in study design, data collection and analysis, decision to publish, or preparation of the manuscript” has been included in the manuscript.

3. Thank you for stating the following financial disclosure: “CeNAT Scholarship Program”

As mentioned in previous point, the Role of Funder was for the financial support awarded to the student during the research period. Therefore, the funders had no role in study design, data collection and analysis, decision to publish, or preparation of the manuscript. This state has been added in the cover letter.

4. In the online submission form, you indicated that “Data will be made available on request.”

All PLOS journals now require all data underlying the findings described in their manuscript to be freely available to other researchers, either 1. In a public repository, 2. Within the manuscript itself, or 3. Uploaded as supplementary information. This policy applies to all data except where public deposition would breach compliance with the protocol approved by your research ethics board. If your data cannot be made publicly available for ethical or legal reasons (e.g., public availability would compromise patient privacy), please explain your reasons on resubmission and your exemption request will be escalated for approval.

We have revised the data and the manuscript and determined that relevant data are within the main manuscript and in the Supplementary material.

The reference list has been checked to ensure it is complete and correct. No changes had to be done.

Reviewer #1:

After reviewing the paper, I am pleased to report that I consider the article is well written and presents a solid piece of research. However, I have identified some areas that could be improved to further strengthen the quality of the document.

Force units

One of the main observations is that in some sections the International System (SI) is not used for force units. I suggest you review and update these units to conform to international standards. This will help improve the coherence and clarity of the manuscript.

Thank you for the observation. All force units were converted to International System (SI) units.

Use of the % sign

Additionally, I have noticed that in some cases the % sign is not separated from the number by a single space. I recommend that you review and correct these cases to maintain a consistent format throughout the text.

Thank you for pointing this out. The percentage sign (%) has been separated from the number by a single space throughout the manuscript. 

Viewing the figures

On the other hand, I have observed that the difference between the dark colors in some of the figures could be further reinforced using different types of plotting. This would make it easier to view and understand the data presented.

Thank you for your valuable observation. The figures have been adjusted accordingly. We hope that this modification meets the intention of making the data easy of viewing and understanding. However, if you have a specific suggestion for further improvement, we would greatly appreciate your input. In addition, we have marked the corresponding mixture and/or sampling in each figure. 

Filter Information

I would also like to point out that when the use of filters is mentioned in the text, no information is given about the specific material of the filters used. This detail may be crucial for the reproducibility of future research based on this study. I suggest you include this information in the corresponding section.

Thanks for counting on me 

Thank you for notices this. The information about the material and other specifications of the filters have been included in the corresponding sections.

---

## [Editor Report · Decision Letter 1]

20 Nov 2024

PONE-D-24-20013R1Formulation and Evaluation of Atorvastatin Calcium Trihydrate Form I TabletsPLOS ONE

Dear Dr. Araya-Sibaja,

Thank you for submitting your manuscript to PLOS ONE. After careful consideration, we feel that it has merit but does not fully meet PLOS ONE’s publication criteria as it currently stands. Therefore, we invite you to submit a revised version of the manuscript that addresses the points raised during the review process. **It is recommended to check style and formatting errors, and correct in light of journal guidelines.**

We look forward to receiving your revised manuscript.

Kind regards,

Amjad Khan, Pharm-D; PhD

Academic Editor

PLOS ONE

**Journal Requirements:**

**Additional Editor Comments:**

Comments

• Text style and format (e.g., P#6, L#117) is not uniform and needs to be checked and corrected throughout the manuscript

• P#6, L#129, There should be space between value and its unit (e.g., 10 mg), except percentage (10%). This point was also raised during previous round of the review. It needs to check and correct throughout the manuscript

• Shift content of the section “Drug content Determination” with the Section “Assay” on P#6

• During compatibility study, was the effect of stress environmental conditions checked? Compatibility cannot be predicted from analysis of pure excipients and dosage form. Sample preparation is not in accordance with the reported literature like 0.1007/s11094-022-02556-8

• On P#13, it is stated that “Sampling points were chosen according to conditions considered critical known as process-induced transformations (PITs) for an API.” Conditions critical for PITs need to be mentioned.

• General things e.g. P#14, L# 298 – 300, are added in “Results and Discussion” section which should be removed. Discussion should be precise to the results. Whole of the manuscript should be reviewed and general statements should be deleted.

• Conclusion is too lengthy. It should be concise and based the experimental results

• In previous round of review, the reviewers suggested to evaluate the list of references which was not addressed, properly. Modify the list of references and add some relevant references like 10.1124/mol.117.110650, 10.1007/s11030-018-9839-y, 10.3390/toxins14120833 and 10.1109/TR.2023.3336330

The manuscript can be accepted for publication, subjected to the incorporation of suggested changes

 **********While revising your submission, please upload your figure files to the Preflight Analysis and Conversion Engine (PACE) digital diagnostic tool, https://pacev2.apexcovantage.com/. PACE helps ensure that figures meet PLOS requirements. To use PACE, you must first register as a user. Registration is free. Then, login and navigate to the UPLOAD tab, where you will find detailed instructions on how to use the tool. If you encounter any issues or have any questions when using PACE, please email PLOS at figures@plos.org. Please note that Supporting Information files do not need this step.

---

## [Author Response · Author response to Decision Letter 1]

18 Dec 2024

The second round contains only editor comments, which have been incorporated into the cover letter.

---

## [Editor Report · Decision Letter 2]

30 Dec 2024

Formulation and Evaluation of Atorvastatin Calcium Trihydrate Form I Tablets

PONE-D-24-20013R2

Dear Dr. Araya-Sibaja,

We’re pleased to inform you that your manuscript has been judged scientifically suitable for publication and will be formally accepted for publication once it meets all outstanding technical requirements.

Kind regards,

Amjad Khan, Pharm-D; PhD

Academic Editor

PLOS ONE
---

## [Editor Report · Acceptance letter]

15 Jan 2025

PONE-D-24-20013R2 

PLOS ONE

Dear Dr. Araya-Sibaja, 

I'm pleased to inform you that your manuscript has been deemed suitable for publication in PLOS ONE. Congratulations! Your manuscript is now being handed over to our production team.

Kind regards, 

on behalf of

Dr. Amjad Khan 

Academic Editor

PLOS ONE